## [Transparent Peer Review file · Nature Communications]

***Acinetobacter baumannii*'s lifestyle includes soil-dwelling colonization of decaying plant material and airborne spread**

Corresponding Author: Dr Gottfried Wilharm

Version 0:

Reviewer comments:

Reviewer #1

(Remarks to the Author)

Summary

The authors conduct what might be the most comprehensive study about isolates of *A. baumannii* from non-human sources, which is very welcome. The authors collected 1450+ non-redundant isolates from many different sources and sequenced 401 genomes. This is a huge amount of work both in terms of data and analyses, which provides novel and interesting insights into the ecology (in particular the habitat) of *A. baumannii* and the importance of the non-human populations of this species. However, some important issues should be addressed in a revised version of the manuscript. There are two relevant technical aspects the authors want to pay attention to (see major comments 1 and 2), other two of background context (major comments 3 and 4) and some non-major issues (final major comment and minor comments) about the structure and tone of the manuscript.

Major comments

1. There are some important technical details regarding the genomes newly sequenced. Please, run CheckM on your genomes and report the completeness and contamination values. Also, please add to Supplementary Table 4 the sequencing statistics (N50, coverage, number of genes, etc.) of the new genomes. These two aspects are relevant to have an idea of the quality of genomes sequenced by the authors. In the results, you want to add one or two sentences about the quality of the genomes sequenced (sequencing statistics, completeness and contamination).
2. Maybe the most salient issue is with the molecular dating analysis. An essential aspect of any molecular dating analysis is to determine if the data has a temporal signal (the mol dating analysis is valid only when there's such a signal). Did the authors check if their data has a temporal signal? They want to do that using TemEst or any similar program. Please report the value of the regression of dates of sampling against the root-to-tip distances. Secondly, why did the authors choose a strict molecular clock? A strict mol clock is not very realistic, biologically speaking. A relaxed molecular clock model is a more sensible option. Third, some information is missing regarding the setting of the substitution model. Please mention the DNA substitution model used for the analysis and other parameters (Proportion of Invariant sites, Gamma category counts, etc.). Finally, but as relevant as the first point, ideally the analysis should be run twice or more to evaluate consistency (similar values should be got for the time to the most recent ancestor (tMRCA) in both runs). Please evaluate the trace plot for the tree likelihood and check if there is good mixing in the tree likelihood. Also, please make sure the effective sampling size of tMRCA is higher than 200 in both runs. The confidence of the mol dating analysis is very important given the statement made by the authors that early adaptive radiation in *A. baumannii* occurred around the Neolithic. Of note, even if there is a weak or lousy temporal signal (and thus the Mol dating analysis is not reliable), I think the rest of the data and analyses in this study are very valuable.
3. There are a couple of recent phylogenomic studies that have studied non-human *A. baumannii* isolates that are relevant to the subject of this paper. A recent genomic and phenotypic study found *A. baumannii* in agricultural surface water, urban streams, and even milk tanks (ref A below). Another recent study found a significant diversity of novel clones of *A. baumannii* in grass (ref B). It would be interesting to know if the OXA-51 allelic variants found in ref B are similar to the OXA-51 variants

found by the authors in their soil and plant isolates. This could indicate that certain clades are preferentially found in soil/plant habitats

Ref A

<https://pubmed.ncbi.nlm.nih.gov/38440986/>

Ref B

<https://pubmed.ncbi.nlm.nih.gov/37439781/>

4. Given that most molecular epidemiology studies of *A. baumannii* use MLST as the standard approach, please state how many new STs (either new combinations of known alleles or even new allelic variants for some loci) were found as per both Pasteur and Oxford MLST schemes. Also, considering the 125 novel OXA-51 variants (line 306 and listed in Supp Table 3) please state the source of the isolates (soil, bird, etc.) and briefly mention which source has most of the novel variants.

5. Finally, some edits can enhance the manuscript. For one, the abstract could be more focused. Also, although the discussion is a great debate on many aspects of the ecology (and even evolution) of *A. baumannii*, it is a bit of a whopper. It feels rather long and some points, although very interesting, are not directly connected to the findings of the paper. For instance, the section about the comparison of the population structure of *A. baumannii* and *P. aeruginosa*. Regarding the methods, the sections "garden compost" and "Collection of rodents, rats and shrews" can be included in the first section "Sample collection and processing".

Minor comments

I think the data linking *A. baumannii* dispersion and fungi are interesting but not conclusive. Maybe the authors want to tone down this a little bit.

Line 63: instead of "around 400" please give the actual number.

Lines 66-68: maybe you want to mention how many different types of sources you sampled.

Line 99: provide the actual number of genomes sequenced.

Line 141-143: This a clear case of reverse Zoonosis, If I were you, I would say explicitly that.

Lines 262-266: how many of those OXA-51 variants were novel?

Line 330: "unique genes" can be misleading; "gene families" or "homologous groups" are better options

Line 624: issue is misspelled.

Reviewer #2

(Remarks to the Author)

"The manuscript, titled "On the ecology of *Acinetobacter baumannii* – jet stream rider and opportunist by nature," aims to describe the ecology of *A. baumannii* by analyzing 1,450 isolates, of which approximately 400 were genome sequenced. It is an interesting and well-written manuscript; however, several points need to be addressed before it is ready for publication.

Some general and specific aims are:

- My main concern is the claim that this work has resolved the ecology of *A. baumannii* by analysing around 1500 strains, of which approximately 400 were sequenced. Given the widespread distribution of *A. baumannii* species in the environment, resolving the ecology of *Acinetobacter baumannii* requires a geographically evenly distributed sampling approach, as there might be several additional hidden niches in the environment, including other animals, etc., that play a pivotal role as reservoirs of strains. I believe the conclusions drawn are biased towards the samples available for this study (i.e. storks and specific geographical regions). Conclusions are valid for the regions studied NOT at the global level. I am concerned because even including the publicly available GenBank genomes also does not normalise the sample biases as only a tiny fraction of the genomes available in GenBank are from environmental samples. I think to address this, the tone and working around this needs to be modified throughout the text.

- Lines 112-115: Not sure about this section of the conclusion, "Taken together, the intrinsic resistance endowment and potential to acquire antibiotic resistance can be explained by coevolution with antibiotic-producing fungi and other microorganisms within the soil," as I think it includes a lot of generalisations. For example, parts of the widespread antimicrobial genomic islands found in *A. baumannii* international clones have been shown to have been acquired from other clinically relevant bacteria such as *Salmonella* and *Enterobacterales*. Overall, I believe this study does not include substantial evidence to support this claim.

- Lines 138-140: Please include the link in the reference list."

- Line 158: Please clarify why the smaller scale? Was the sample size determined through statistical methods? Please clarify!
- Lines 169-172: Please include some numbers (e.g. colonisation, mortality rate etc.).
- Line 171: Did you identify increased white blood cell counts in all colonised individuals? Please clarify.
- Line 176: In the section "We collected pellets from selected nests", how did you ensure that the samples are genuinely from pellets and nothing else (e.g. egg shells OR feathers)? Please clarify!
- Line 178: This is an interesting observation, the fact that no pellets was positive earlier than May-April, could this be a correlation with the rain season in the area and that pellets might have washed away? Could you please elaborate on this?
- Line 186: Could this be due to the seasonal conditions?
- Line 248: Please specify which regions?
- Line 317-318: Please include the sequence type number (ST#) representing each of the international clones. (e.g. OXA-66, ST2 etc.).
- the blaOXA-51-like and blaADC are considered part of the core genome in *A. baumannii*, so can you elaborate on the sequence type(s) of those that lack these genes? And is there a correlation between the lack of both genes (i.e. when blaADC is absent blaOXA-51-like is also absent)?
- Lines 381-389: Please include the sequence type(s) of those with AMR genes. This is important given that all the significance of *A. baumannii* is in the context of AMR.
- Line 402: Please be specific here, how many isolates were recovered? Was a sufficient number of strains recovered to support the claim (i.e. airborne spread)?
- Line 412: in all samples tested? Please clarify!
- Line 436: This is not accurate, several publications including genome sequence data have documented the isolation of IC1 (ST1) from natural environments. Please fix this.
- Lines 441-461: I can see a lot of merits in comparing the population structures of *A. baumannii* and *Pseudomonas aeruginosa*, however, it seems very out of place in the context of this study given the completely different evolutionary stories and global/clonal expansions that these two genera have experienced. Please remove this section.
- The discussion is very long, the manuscript will benefit more from a more concise discussion.

Reviewer #3

(Remarks to the Author)

Dear Authors,

I enjoyed and spent some time reviewing your manuscript 'On the ecology of *Acinetobacter baumannii* – jet stream rider and opportunist by nature'

Your manuscript is conspicuous, well clearly presented and participated and it contributes to greater understanding the ecology and evolution of *A. baumannii* in natural habitats, that to date is really poorly understood outside the hospital.

My compliments for the supplementary datasets that are rich in content and very useful for data visualization and interpretations.

All the comments and suggestions I have made are of a minor nature and I would be happy for the manuscript to be accepted for publication in Nature Communication once these have been appropriately addressed point by point as follows:

- Why authors identified white storks as a model system to study the ecology of *A. baumannii*? Why not other wild or synanthropic birds?
- Considering the huge epidemiological study, I wonder if the authors also found other pathogens of interest from the samples analyzed.
- Regarding the Diversity in terms of OXA-51 variants collected from Poland and Germany and the Phylogenomics-based representation of the diversity, I agree with the choice of at least one strain representing each OXA variant for whole genome sequencing.
- Very useful and well done the 'Phylogenetic distribution of strains carrying selected variants similar to OXA-51' illustrated in Suppl. Fig. S14 and deepened in the microreaction project

Regarding 'Compost, soil, rhizosphere and the forest paradox' (line 257), really interesting the sampling of the soil originating from a single compost across a period of seven months that yielded 86 isolates with 20 different variants of the OXA-51 family, and earthworms within the same compost that yielded 10 different variants of OXA-51. Instead, (line 298-301) although very interesting as data, considering the low number of samples examined, I would go with more moderation in stating that *A. baumannii* can thrive along streams and rivers outside of forests. The decomposition of plant material by fungi would seem to lay the foundation for the proliferation of *A. baumannii*. You could clearly assume that decomposition of plant material by fungi would seem to appear to set the stage for proliferation of *A. baumannii*.

- Line 398: Regarding the Colonization of compost material and linkages to the fungal world, the experiment is not so clear to me. How many tubes did you mounted? How many trials have you conducted? Please specify
- Line 845: Collection of rodents, rats and shrews. How many cat-captured rodents have you collected? Particularly, can you describe in detail the capture procedure and whether there are laws and permits (especially for rats) in the country of collection (in addition to those already mentioned) about the associated biological risks.

FIGURES

- Figures 3, 4, 5 are obviously simply representative and at first glance difficult to understand especially because of the amount of isolates represented graphically. The link <https://microreact.org> clearly describes and represents this. But to make the graphic part of the figures, especially 4 and 5, more immediate and understandable, please divide each of the figures into A and B and briefly describe the individual parts before directing the reader to the link.
- Please enlarge the central part of Fig. S1 'Spatial correlogram' to make it more readable
- I suggest putting letters A, B, C... for each Supplementary Figure consisting of two or more figures and where the legend is not properly specified, to make them even clearer (E.g. Suppl. Fig. S18)

Hope my review could help you to improve your work.

Regards

Reviewer #4

(Remarks to the Author)

In this study, Wilharm et al. investigated multiple aspects of ecology and distribution of *Acinetobacter baumannii*. Authors have dedicated a quite a bit of resources for this study and provide some novel insights into the ecology and evolution of *A. baumannii* complex. However, I find the current form of the manuscript, presentation of the results and the analyses do not justify the effort put into this study and data generated by the authors. Also, there are few concerns related to methodology of this study. Below I highlight my major concerns and comments.

1. Structure and the organization of the manuscript: The current written style of this manuscript includes a collage of mini experiments, at times seems very fragmented.
 - a. The introduction itself is not giving a proper background to justify why it is important to conduct the study that authors have done, and I don't think "we intended to unravel the ecology of these bacteria in their natural habitats" is a good argument. I would suggest re-writing a proper introduction highlighting the thought process and major gaps that authors are anticipating on filling with their work. Also, it is a bit weird to see the first citation in the introduction is the 90th citation in the reference list.
 - b. Detailed results related to isolation information of the *A. baumannii* can go into the supplementary and just provide a

summary of what was isolated.

c. Authors need to analyze the genomes bit more thoroughly. This is the most interesting dataset of the study and there are many analyses that authors can do to identify the ecology and evolution of this species complex (see my major point 3).

d. Supplementary: There are two supplementary files and one includes the figures and the other includes a mix of things which also have some interesting results on GIS based analyses. I think these two documents need to be merged (one of this document even have a figure 1). I also think the GIS analyses need to be highlighted in the main text results and to show whether the number of isolates have an association with the proximity to human activity. This is what I saw with your transect study, where you only isolate *A. baumannii* after the village (Fig S12).

e. Discussion is extremely long and highly speculative, while making huge statements without adequate support. I think authors need to first define their objectives and make the discussion more streamlined and avoid jargon.

2. Uneven sampling:

a. Authors have done an amazing job with sampling White storks in Germany and Poland. Then they have attempted to sample other bird species (grey-herons, kestrels and lack storks), which have a very small number of individuals compared to the white stork samples and only sampled the cloaca of the birds. This raises the question whether this uneven sampling led to the observed lack of isolation of *A. baumannii* from other bird species. Second, in White storks it seems like *A. baumannii* is isolated more from choanae and for other species authors took cloacal samples. Again, this makes me wonder if authors sampled choanae in other birds whether they would have gotten different set of results.

b. Autoclaved compost test: This was an interesting approach but why did you only do this in a garden. I think it would make more sense to conduct this experiment both in the garden setting, semi-natural area and in the forest environment. This way you will have the opportunity to test your hypotheses related to airborne nature of dispersion of *A. baumannii* away from human settlements.

3. Genome analyses: I feel like this is the most important dataset of the study and authors just looked at the surface of this dataset.

a. It would be good to show genes/gene families/SNP differences between environmental isolates and clinical isolates in a figure (it should be a main figure).

b. Given that authors have identify clades of environmental, animal and clinical isolates I think authors should conduct positive selection analyses to investigate if there are certain genes under positive selection when *A. baumannii* become a hospital strain. Moreover, this will allow you to identify if there are particular genes that show converged positive selection when strains become clinical isolates across the clades.

c. Molecular clock analysis: What *Acinetobacter* species did the authors used as the outgroup when doing this analysis? Also given the diversity and the geographic distribution of the genomes you have, you should conduct ancestral state reconstruction to see if you can point our a potential origin of the radiation of this species clade.

4. Tracing the food sources: I think the proper way to do this is first to conduct metabarcoding of the pallet material. Then identify food items common in pallets with *A. baumannii* isolates and then collect the wild individuals (slikewhat you did with earthworms) and then see if those wild diet items carry *A. baumannii*.

5. I also wonder why authors did not conduct whole microbiome sequencing of their samples to confirm that isolation actually represent the presence of *A. baumannii*. This is because the isolation alone might not give the full picture of the presence of *A. baumannii* in a given sample. Have you done any preliminary work to make sure you always identify *A. baumannii* from samples when the bacteria are present?

Minor comments

L127: Citations are weird. Please double check.

L154: Is "average rate of 25%" mean they were isolated from 25% of samples. In that case isn't it the prevalence?

L164: Terminology for birds is "Cloacal" not "Rectal".

L233: Can you split the isolate numbers of the bacteria according to different earthworm genera?

L254-256: I think this is an issue of sampling depths and sampling types.

Oxa-51 variants: What is your cutoff value to call a strain as a different variant? Is it 1bp difference? Also, what is the length of this segment?

L330-331: Can you summaries the geographic distribution of your genomes? This will help readers to easily see if there are geographic biases in the isolates you used in this study.

L413: Is the incubation temperature you used natural (e.g., environmental temperature)? Also, if the adherence of *A. baumannii* inhibit the germination of fungal spores, can that also impact other properties of the spores (like the dispersal ability)?

441-461: Very weird section to start your discussion with, given that the study was not a comparative study between *P. aeruginosa* and *A. baumannii*.

462-501: This part is also very speculative, and I struggle to find how this relate to authors findings, except for that authors

show the ability of the bacteria to adhere to fungi. I think if authors want to discuss more "Evidence for interrelatedness of *A. baumannii* life cycle with the fungal world" they need to conduct more experiment to properly show the claims they make here.

L509: Basidiomycota is a very broad and diverse division of fungi and don't think using this is a good argument here.

L541-585: I am not sure how your findings support *A. baumannii* uses air currents to travel 1000s of kms between continents. Maybe I missed something here?

L559-560: This I don't understand. There seem to be an association with your isolates and presence of humans and then here you say "All in all, there is no evidence that the observed global pattern of distribution of environmental *A. baumannii* is caused by human activity". You have data to test this!

L660-668: This section seems like it is repeating things. Please add this section to other corresponding sections.

L809: Did the incubation happened at room temperature?

L822-828: Results for this methods section is not in the "Results" section. I think it is very interesting and should be in the main paper.

Figures: Please provide better legends for the figures. The interactive link is cool but you need to provide enough information these figure legends to guide reads (figures and legends should be able to stand alone). Also please provide information on the color codes associated with phylogenies.

Version 1:

Reviewer comments:

Reviewer #1

(Remarks to the Author)

The authors have addressed my comments. Congratulations to Gottfried and colleagues for all their hard work.

Reviewer #2

(Remarks to the Author)

N/A

Reviewer #3

(Remarks to the Author)

Dear Authors,

I noted all my suggestions and those of the other reviewers have been well accepted and incorporated into the text.

The manuscript is now certainly more robust and deserves to be published without further revisions, in my opinion.

Good luck and congratulations

Kind regards

Reviewer #4

(Remarks to the Author)

This study by Wilharm et al., provides an interesting dataset on *A. baumannii*, but I do not think authors are doing enough justification for this amazing dataset they have (i.e, genomes) produced here. This version of the manuscript still provides a simple story on where and how to isolate these bacteria (which is very interesting) and present some hypothesis on how this bacteria species might spread around the world. However, I still think the major value of this study are the new genomes, thus, I still think more comparative analyses of these genomes are necessary to properly understand the underlying genomic features that might drive *A. baumannii* to become clinically relevant strains (which should also play the center role).

Given that sampling is bias toward Germany and Poland, I don't think (even with the great number of isolates produce by this study), authors have enough evidence/data to give a global overview of this species complex. What you have are amazing set of genomes that you can answer many more interesting questions related to how *A. baumannii* become virulent in hospital settings. Such a story will fit well with the aims and interest the broad readership of this journal.

Secondly, I still don't agree with authors proposed mechanism of the spread of this species complex, mainly because the proposed mechanism is based on many non-tested assumptions or over extrapolating some findings. Also, authors tend to ignore potential steppingstone colonization methods, spread by other migrant species (not just storks), and potential combination of many methods including the airborne spread as authors suggested. I think, to receive a better outcome from this amazing work, the scope of the study should change and generate more specific research questions (related to the evolution, and genetic underpinnings of virulence) instead of just saying you are trying to "improve the understanding of the ecology of *A. baumannii* in natural habitats...".

Point-by-point response

REVIEWER COMMENTS

Reviewer #1:

Summary

The authors conduct what might be the most comprehensive study about isolates of *A. baumannii* from non-human sources, which is very welcome. The authors collected 1450+ non-redundant isolates from many different sources and sequenced 401 genomes. This is a huge amount of work both in terms of data and analyses, which provides novel and interesting insights into the ecology (in particular the habitat) of *A. baumannii* and the importance of the non-human populations of this species. However, some important issues should be addressed in a revised version of the manuscript. There are two relevant technical aspects the authors want to pay attention to (see major comments 1 and 2), other two of background context (major comments 3 and 4) and some non-major issues (final major comment and minor comments) about the structure and tone of the manuscript.

Major comments

1. There are some important technical details regarding the genomes newly sequenced. Please, run CheckM on your genomes and report the completeness and contamination values. Also, please add to Supplementary Table 4 the sequencing statistics (N50, coverage, number of genes, etc.) of the new genomes. These two aspects are relevant to have an idea of the quality of genomes sequenced by the authors. In the results, you want to add one or two sentences about the quality of the genomes sequenced (sequencing statistics, completeness and contamination).

We fully agree with the reviewer that information regarding genome quality is of essence to evaluate any potential impact on downstream analyses. To provide this, we added a section on genome assembly assessment (lines 672-679) using the tools BMap and checkM. A majority of the assessed genomes was of high quality (high completeness, low contamination) and thus suitable for the presented downstream analyses. A full overview of the assessed statistics is listed in new Suppl. Table S9. Completeness and contamination values have also been introduced into Suppl. Table S4 as requested.

2. Maybe the most salient issue is with the molecular dating analysis. An essential aspect of any molecular dating analysis is to determine if the data has a temporal signal (the mol dating analysis is valid only when there's such a signal). Did the authors check if their data has a temporal signal? They want to do that using TemEst or any similar program. Please report the value of the regression of dates of sampling against the root-to-tip distances. Secondly, why did the authors choose a strict molecular clock? A strict mol clock is not very realistic, biologically speaking. A relaxed molecular clock model is a more sensible option. Third, some information is missing regarding the setting of the substitution model. Please mention the DNA substitution model used for the analysis and other parameters (Proportion of Invariant sites, Gamma category counts, etc.). Finally, but as relevant as the first point, ideally the analysis should be run twice or more to evaluate consistency (similar values should be got for the time to the most recent ancestor (tMRCA) in both runs). Please evaluate the trace plot for the tree likelihood and check if there is good mixing in the tree likelihood. Also, please make sure the effective sampling size of tMRCA is higher than 200 in both runs. The confidence of the mol dating analysis is very important given the statement made by the authors that early adaptive radiation in *A. baumannii* occurred around the Neolithic. Of note, even if there is a weak or lousy temporal signal (and thus the Mol dating analysis is not reliable), I think the rest of the data and analyses in this study are very valuable.

Following the suggestion of the reviewer, we performed temporal signal analysis on the data through TempEST. As the reviewer correctly estimated, the detected temporal signal of the data set was overall rather weak, potentially impacting the accuracy of the time-based analysis. However, we believe that the novelty of the results outweighs the initially detected weak signal. Thus, to compensate for the lowered temporal signal, we based our further parameter selection on previously published methods utilized for the research on international clone 1, including the decision to adapt a strict molecular clock and an GTR-gamma DNA substitution model (Holt et al., 2016). To address the reviewer's question, we have added a section containing details of the temporal signal analysis (lines 718-726). We consequently toned down regarding our claims based on the molecular clock but kept the main conception considering the overall evidence. Moreover, we critically mentioned the weaknesses in the "Limitations" section (lines 627-631).

3. There are a couple of recent phylogenomic studies that have studied non-human *A. baumannii* isolates that are relevant to the subject of this paper. A recent genomic and phenotypic study found *A. baumannii* in agricultural surface water, urban streams, and even milk tanks (ref A below). Another recent study found a significant diversity of novel clones of *A. baumannii* in grass (ref B). It would be interesting to know if the OXA-51 allelic variants found in ref B are similar to the OXA-51 variants found by the authors in their soil and plant isolates. This could indicate that certain clades are preferentially found in soil/plant habitats

Ref A

<https://pubmed.ncbi.nlm.nih.gov/38440986/>

Ref B

<https://pubmed.ncbi.nlm.nih.gov/37439781/>

We have compared our dataset with that of the two recent publications. On the basis of MLST which is more reliable compared to OXA-51-like variants alone, we identified 3 STs of Ref. A that were also represented in our dataset (2 stork isolates, 1 earthworm isolate), and 1 ST of Ref. B represented in our dataset (1 stork isolate). We consider this database insufficient for a discussion on clades preferentially associated with soil/plant habitats even if we assume the stork isolates to stem from the same habitats. However, we have briefly mentioned and cited the references to further illustrate the One Health relevance (see lines 617-620).

4. Given that most molecular epidemiology studies of *A. baumannii* use MLST as the standard approach, please state how many new STs (either new combinations of known alleles or even new allelic variants for some loci) were found as per both Pasteur and Oxford MLST schemes. Also, considering the 125 novel OXA-51 variants (line 306 and listed in Supp Table 3) please state the source of the isolates (soil, bird, etc.) and briefly mention which source has most of the novel variants.

The requested information on the number of new sequence types (ST) has now been added to the manuscript (see lines 681-683) and the Suppl. Table 3 has been amended to give an overview on the sources of isolates with novel OXA-51 variants. Most of them were associated with isolates from white stork nestling as has been now stated in the manuscript (see line 305).

5. Finally, some edits can enhance the manuscript. For one, the abstract could be more focused. Also, although the discussion is a great debate on many aspects of the ecology (and even evolution) of *A. baumannii*, it is a bit of a whopper. It feels rather long and some points, although very interesting, are not directly connected to the findings of the paper. For instance, the section about the comparison of the population structure of *A. baumannii* and *P. aeruginosa*. Regarding the methods, the sections “garden compost” and “Collection of rodents, rats and shrews” can be included in the first section “Sample collection and processing”.

The abstract is now considerably condensed (see lines 84-103). As requested, we have removed the comparison of population structures of *A. baumannii* and *P. aeruginosa* and further streamlined the discussion outlined above. We changed the Methods section according to the reviewer’s advice (see lines 645-656).

Minor comments

I think the data linking *A. baumannii* dispersion and fungi are interesting but not conclusive. Maybe the authors want to tone down this a little bit.

We have toned down in these parts on the request of several reviewers. Moreover, we have restructured the corresponding discussion sections to improve the presentation.

Line 63: instead of “around 400” please give the actual number.

Corrected: 401 (see line 63)

Lines 66-68: maybe you want to mention how many different types of sources you sampled.

We have refrained from following this advice to save space and because of the ambiguity to define what “different” sources are.

Line 99: provide the actual number of genomes sequenced.

Corrected: 401 (see line 94)

Line 141-143: This a clear case of reverse Zoonosis, If I were you, I would say explicitly that.

Done, thank you! (see line 128)

Lines 262-266: how many of those OXA-51 variants were novel?

122 were novel (see line 303)

Line 330: “unique genes” can be misleading; “gene families” or “homologous groups” are better options

Changed to “gene families” (see line 329 and also abstract lines 95-96)

Line 624: issue is misspelled.

Omitted in the course of streamlining the discussion!

Reviewer #2:

"The manuscript, titled "On the ecology of *Acinetobacter baumannii* – jet stream rider and opportunist by nature," aims to describe the ecology of *A. baumannii* by analyzing 1,450 isolates, of which approximately 400 were genome sequenced. It is an interesting and well-written manuscript; however, several points need to be addressed before it is ready for publication.

Some general and specific aims are:

- My main concern is the claim that this work has resolved the ecology of *A. baumannii* by analysing around 1500 strains, of which approximately 400 were sequenced. Given the widespread distribution of *A. baumannii* species in the environment, resolving the ecology of *Acinetobacter baumannii* requires a geographically evenly distributed sampling approach, as there might be several additional hidden niches in the environment, including other animals, etc., that play a pivotal role as reservoirs of strains. I believe the conclusions drawn are biased towards the samples available for this study (i.e. storks and specific geographical regions). Conclusions are valid for the regions studied NOT at the global level. I am concerned because even including the publicly available GenBank genomes also does not normalise the sample biases as only a tiny fraction of the genomes available in GenBank are from environmental samples. I think to address this, the tone and working around this needs to be modified throughout the text.

Thank you! These comments helped us a lot to clarify statements and claims. However, we felt that our previous statements already made clear, that our study has not completely “resolved the ecology of *A. baumannii*”. However, now we more explicitly pointed out limitations and biases and toned down on claims and conclusions, also in line with concerns of other reviewers (see below). But please also acknowledge our previous statements, e.g. in the “Limitations” section where we already had written “Although we could demonstrate worldwide spread of lineages isolated in Poland and Germany, not all lineages necessarily spread worldwide. Ecology of *A. baumannii* in other climates and geographic regions may differ considerably.” Please see also another previous statement “Although our sample setting, essentially restricted to samples from Poland and Germany, represents a significant portion of the worldwide known diversity, evidently, our picture remains incomplete.” (now in lines 446-448”).

- Lines 112-115: Not sure about this section of the conclusion, "Taken together, the intrinsic resistance endowment and potential to acquire antibiotic resistance can be explained by coevolution with antibiotic-producing fungi and other microorganisms within the soil," as I think it includes a lot of generalisations. For example, parts of the widespread antimicrobial genomic islands found in *A. baumannii* international clones have been shown to have been acquired from other clinically relevant bacteria such as *Salmonella* and *Enterobacterales*. Overall, I believe this study does not include substantial evidence to support this claim.

Thank you! We have rewritten parts of the Summary and toned down on claims. Clearly, acquired resistance genes play a major role in the clinical context. The natural endowment of the species, however, already includes lots of resistance genes and efflux pumps conferring resistance to chloramphenicol and various beta-lactams amongst others. Moreover, the permeability of the outer membrane is extremely low compared to that of *Enterobacterales* for instance. This is what was meant by “intrinsic resistance endowment” which clearly evolved in natural environments, and most likely, particularly in soil habitats.

- Lines 138-140: Please include the link in the reference list."

Done! (see line 126)

- Line 158: Please clarify why the smaller scale? Was the sample size determined through statistical methods?

Due to the large effort, the sample size was kept small given that we simply wanted to prove for Germany the same tendency as observed for Poland. The determination of the scale was without statistical means. We have limited the statement on differences in the ecology of bacteria and/or storks to a comparison of Poland and Spain to account for the reviewer's concern of a small sample size of storks from Germany (see line 158).

Lines 169-172: Please include some numbers (e.g. colonisation, mortality rate etc.).

We have omitted the statement on mortality because of low statistical power of the observations. Years with high colonization rates with *A. baumannii* among nestlings apparently did not lower the breeding success.

- Line 171: Did you identify increased white blood cell counts in all colonised individuals? Please clarify.

No, the increase of white blood cell count was not identified in the complete set of colonized individuals, but the difference to non-colonised individuals was highly significant (see Suppl. Table S2).

- Line 176: In the section "We collected pellets from selected nests", how did you ensure that the samples are genuinely from pellets and nothing else (e.g. egg shells OR feathers)? Please clarify.

Thank you, this is a very important point. The pellet samples for enrichment cultures were taken from the center of the pellet with sterile tweezers to exclude enrichment of contaminations from the outer surface of the pellet. We have now also specified this in the manuscript (see lines 643-644). To illustrate that *A. baumannii* is really found inside of the pellets, we had also provided Suppl. Fig. S4. When breaking open a pellet, the interior can be directly stamped on a CHROMagar Acinetobacter selective agar thus demonstrating the presence of *A. baumannii* inside the pellet.

- Line 178: This is an interesting observation, the fact that no pellets was positive earlier than May-April, could this be a correlation with the rain season in the area and that pellets might have washed away? Could you please elaborate on this?

This observation is perfectly matched by our later analyses on soil samples. There is a clear seasonality and it appears to be a temperature dependence in the first place. In compost, where temperatures are higher, the season for *A. baumannii* is longer. As mentioned in the discussion, seasonality has been also described in cattle. More important, seasonality is also observed in the clinical setting, but only for non-MDR isolates which apparently enter the hospital at higher rates during the summer season (see lines 453-463).

- Line 186: Could this be due to the seasonal conditions?

Yes. Apparently, adult storks are not colonized by *A. baumannii* when they arrive from their wintering grounds, otherwise we should have detected them in pellet material in February–March. Only after the season for *A. baumannii* has started in the soil, we also see increasing numbers of positive pellets. This is why your concern regarding regional and global statements is of course highly relevant. We cannot deduce on what happens in tropical and subtropical climates for example (but please see our “Limitations” section). All the more remarkable it is, that our data collection covers such a large part of the worldwide known diversity. As far as we can see, only a very special dispersal mechanism can explain this.

- Line 248: Please specify which regions?

We now added “Germany” in the manuscript (see line 244), but the specific place “Neuendorf, Schleswig-Holstein” is available through Suppl. Table S4 for the interested reader.

- Line 317-318: Please include the sequence type number (ST#) representing each of the international clones. (e.g. OXA-66, ST2 etc.).

Done! (see lines 316-318)

- the blaOXA-51-like and blaADC are considered part of the core genome in *A. baumannii*, so can you elaborate on the sequence type(s) of those that lack these genes? And is there a correlation between the lack of both genes (i.e. when blaADC is absent blaOXA-51-like is also absent)?

The sequence types scatter for strains lacking either *bla*_{OXA-51-like} or *bla*_{ADC}. We found not a single isolate lacking both genes. A detailed exploration is possible via the microreact tool.

- Lines 381-389: Please include the sequence type(s) of those with AMR genes. This is important given that all the significance of *A. baumannii* is in the context of AMR.

Only 12 isolates of our collection exhibit AMR gene counts above 3. Six of these are human clinical isolates representing ST^{Pas} 2, 106, 1357, 2402, and 2409, four are animal isolates of veterinary and livestock origin representing ST^{Pas} 23, 241 and 866. Only two isolates of environmental and wildlife origin reach a count of 4 AMR genes (ST^{Pas} 23 and 309). This is available via Suppl. Table S4 and now mentioned in lines 380-384.

- Line 402: Please be specific here, how many isolates were recovered? Was a sufficient number of strains recovered to support the claim (i.e. airborne spread)?

We describe here the first piece of evidence which arose from a few preliminary experiments with tubes mounted above the compost. Of these five samples, four were positive for *A. baumannii*. The positive finding stimulated more systematic experiments with sterile plant material deposited in the garden. These deposition experiments were successful in four out of four cases. These numbers have now been provided in the manuscript (see lines 411-415).

- Line 412: in all samples tested? Please clarify.

This kind of experiment has been performed multiple times by three students independently over the years. They all consistently demonstrated the coating of different spores from *Aspergillus* and *Penicillium* species with different strains of *A. baumannii*. The level of coating illustrated in Fig. 7 is representative of the level of coating visualized by bright field and fluorescence microscopy. We shall be happy to provide additional pictures if this is requested.

- Line 436: This is not accurate, several publications including genome sequence data have documented the isolation of IC1 (ST1) from natural environments. Please fix this.

First of all, we noted that IC1, IC2 and IC3 are missing in our own collection of environmental isolates. This is accurate. Secondly, the environmental isolates described by others belonging to IC1-3 typically exhibit signs of escape from hospital or livestock production, e.g. typical resistance profiles, or at least are from environments of ambient human activity. They therefore do not represent original environmental isolates. Unfortunately, we could not identify the genomes mentioned by the reviewer.

- Lines 441-461: I can see a lot of merits in comparing the population structures of *A. baumannii* and *Pseudomonas aeruginosa*, however, it seems very out of place in the context of this study given the completely different evolutionary stories and global/clonal expansions that these two genera have experienced. Please remove this section.

As suggested, we have removed this section!

- The discussion is very long, the manuscript will benefit more from a more concise discussion.

We followed this suggestion and have reduced and streamlined the discussion!

Reviewer #3:

Dear Authors,

I enjoyed and spent some time reviewing your manuscript 'On the ecology of *Acinetobacter baumannii* – jet stream rider and opportunist by nature'

Your manuscript is conspicuous, well clearly presented and participated and it contributes to greater understanding the ecology and evolution of *A. baumannii* in natural habitats, that to date is really poorly understood outside the hospital.

My compliments for the supplementary datasets that are rich in content and very useful for data visualization and interpretations.

Thank you!

All the comments and suggestions I have made are of a minor nature and I would be happy for the manuscript to be accepted for publication in Nature Communication once these have been appropriately addressed point by point as follows:

- Why authors identified white storks as a model system to study the ecology of *A. baumannii*? Why not other wild or synanthropic birds?

As outlined earlier (Wilharm et al., 2017), we started with the hypothesis that the ability of *A. baumannii* to efficiently grow at temperatures above 40°C could point to an adaptation to hosts with core body temperatures around 40°C such as birds. White storks are intensively studied and nestlings regularly ringed on a broad scale. This means that there is no extra sampling campaign necessary. Moreover, the sampling is very easy due to the large size of the nestlings and it is well established. The nestlings show a so-called akinesia that prevents them from falling out of the nest. Further, we considered migratory birds to be particularly interesting due to the possibility of intercontinental transfer between Africa and Europe. All in all, storks were simply the most easily accessible starting point. By chance, it turned out to be a fascinating model system.

- Considering the huge epidemiological study, I wonder if the authors also found other pathogens of interest from the samples analyzed.

For most of the time we used selective media and enrichment methods as outlined in the Materials and Methods section so that we only rarely isolated microorganisms other than *Acinetobacter*. Only in the beginning of our studies we also systematically used non-selective media that indeed revealed a plethora of potential pathogens including *E. coli*, *Klebsiella* and *Pseudomonas*. The results are unpublished except for a few novel species discovered. Having this wealth in mind, we kept a considerable part of our sampling material in the freezer that can be used for future studies.

- Regarding the Diversity in terms of OXA-51 variants collected from Poland and Germany and the Phylogenomics-based representation of the diversity, I agree with the choice of at least one strain representing each OXA variant for whole genome sequencing.

Thank you!

- Very useful and well done the 'Phylogenetic distribution of strains carrying selected variants similar to OXA-51' illustrated in Suppl. Fig. S14 and deepened in the microreaction project

Thank you!

Regarding 'Compost, soil, rhizosphere and the forest paradox' (line 257), really interesting the sampling of the soil originating from a single compost across a period of seven months that yielded 86 isolates with 20 different variants of the OXA-51 family, and earthworms within the same compost that yielded 10 different variants of OXA-51.

Instead, (line 298-301) although very interesting as data, considering the low number of samples examined, I would go with more moderation in stating that *A. baumannii* can thrive along streams and rivers outside of forests.

Thank you! We have attenuated our statements accordingly (see lines 295-297).

The decomposition of plant material by fungi would seem to lay the foundation for the proliferation of *A. baumannii*. You could clearly assume that decomposition of plant material by fungi would seem to appear to set the stage for proliferation of *A. baumannii*.

- Line 398: Regarding the Colonization of compost material and linkages to the fungal world, the experiment is not so clear to me. How many tubes did you mounted? How many trials have you conducted? Please specify

Regarding colonization of compost material, these deposition experiments with sterile material yielded *A. baumannii* in 4 out of 4 cases. Before, we had conducted preliminary experiments in which we mounted 5 tubes above the compost. Of these 5 samples, 4 were positive for *A. baumannii*. These numbers have now been provided in the manuscript (see lines 411-415). The interaction between fungi and *A. baumannii* in vitro have been studied multiple times over the years by different students with consistent reproducibility. We have rewritten parts of the discussion section and also toned down on this topic.

- Line 845: Collection of rodents, rats and shrews. How many cat-captured rodents have you collected? Particularly, can you describe in detail the capture procedure and whether there are laws and permits (especially for rats) in the country of collection (in addition to those already mentioned) about the associated biological risks.

The numbers of cat-captured rodents (n=154) and shrews (n=64) have been detailed in Supplementary Table S1. Regarding cat-captured animals, only dead animals deposited by cats at their respective homes were included. Wild rats were captured by professional pest control operators, and hence no capture permit was required (see referenced work by Raafat et al., 2020). Other small mammals were captured as described in referenced work by Jeske et al., 2021.

FIGURES

- Figures 3, 4, 5 are obviously simply representative and at first glance difficult to understand especially because of the amount of isolates represented graphically. The link <https://microreact.org> clearly describes and represents this. But to make the graphic part of the figures, especially 4 and 5, more immediate and understandable, please divide each of the figures into A and B and briefly describe the individual parts before directing the reader to the link.

Done!

- Please enlarge the central part of Fig. S1 'Spatial correlogram' to make it more readable

Done!

- I suggest putting letters A, B, C... for each Supplementary Figure consisting of two or more figures and where the legend is not properly specified, to make them even clearer (E.g. Suppl. Fig. S18)

Done!

Hope my review could help you to improve your work.

Regards

Thank you again!

Reviewer #4:

In this study, Wilharm et al. investigated multiple aspects of ecology and distribution of *Acinetobacter baumannii*. Authors have dedicated a quite a bit of resources for this study and provide some novel insights into the ecology and evolution of *A. baumannii* complex. However, I find the current form of the manuscript, presentation of the results and the analyses do not justify the

effort put into this study and data generated by the authors. Also, there are few concerns related to methodology of this study. Below I highlight my major concerns and comments.

1. Structure and the organization of the manuscript: The current written style of this manuscript includes a collage of mini experiments, at times seems very fragmented.

a. The introduction itself is not giving a proper background to justify why it is important to conduct the study that authors have done, and I don't think "we intended to unravel the ecology of these bacteria in their natural habitats" is a good argument. I would suggest re-writing a proper introduction highlighting the thought process and major gaps that authors are anticipating on filling with their work. Also, it is a bit weird to see the first citation in the introduction is the 90th citation in the reference list.

We felt that "...to unravel the ecology of these bacteria in their natural habitats" is a good argument, because this piece was missing and we expected it to contribute to our understanding of its tenacity and success in the hospital setting. We have tried to improve the introduction (see lines 133-144). Further, we switched to the reference style of the journal now.

b. Detailed results related to isolation information of the *A. baumannii* can go into the supplementary and just provide a summary of what was isolated.

We have reduced the section as much as possible but also received contrary advice from other reviewers to provide even more information.

c. Authors need to analyze the genomes bit more thoroughly. This is the most interesting dataset of the study and there are many analyses that authors can do to identify the ecology and evolution of this species complex (see my major point 3).

Thank you! We have extended our genome-based analyses. See below our response to major point 3!

d. Supplementary: There are two supplementary files and one includes the figures and the other includes a mix of things which also have some interesting results on GIS based analyses. I think these two documents need to be merged (one of this document even have a figure 1). I also think the GIS analyses need to be highlighted in the main text results and to show whether the number of isolates have an association with the proximity to human activity. This is what I saw with your transect study, where you only isolate *A. baumannii* after the village (Fig S12).

Due to the conflicting advice from other reviewers we kept the GIS analyses in the Supplements but tried to better organize the supplementary material as advised.

e. Discussion is extremely long and highly speculative, while making huge statements without adequate support. I think authors need to first define their objectives and make the discussion more streamlined and avoid jargon.

We have considerably streamlined the discussion and as detailed below our hypothesis on the airborne global spread of *A. baumannii* received significant support from a recent study. We further removed the highly speculative statements mentioned by the reviewer and use of jargon.

2. Uneven sampling:

a. Authors have done an amazing job with sampling White storks in Germany and Poland. Then they have attempted to sample other bird species (grey-herons, kestrels and lack storks), which have a very small number of individuals compared to the white stork samples and only sampled the cloaca of the birds. This raises the question whether this uneven sampling led to the observed lack of isolation of *A. baumannii* from other bird species. Second, in White storks it seems like *A. baumannii* is isolated more from choanae and for other species authors took cloacal samples. Again, this makes me wonder if authors sampled choanae in other birds whether they would have gotten different set of results.

We concur in the assessment of uneven sampling both in terms of numbers and in terms of differences of the sampling materials. We have acknowledged this, now stating "...apparently differ from white stork regarding carriage of *A. baumannii* although this might be influenced by differences in sampling depth and sampling types." (see lines 250-252). However, even if we compare the same kind of sample material, none of the other three organisms reaches the prevalence levels of the white stork.

b. Autoclaved compost test: This was an interesting approach but why did you only do this in a garden. I think it would make more sense to conduct this experiment both in the garden setting, semi-natural area and in the forest environment. This way you will have the opportunity to test your hypotheses related to airborne nature of dispersion of *A. baumannii* away from human settlements.

Definitely, it will be interesting to learn in which environment we see the highest airborne load but this was out of the scope of this study, also because of the enormous effort to obtain permissions. At least, in global support of our hypothesis of airborne dispersal of *A. baumannii*, a very recent paper by Rodó et al. (PMID: 39250672) explicitly reports on the possibility of long-distance transport of pathogens including *A. baumannii* based on tropospheric sampling.

3. Genome analyses: I feel like this is the most important dataset of the study and authors just looked at the surface of this dataset.

a. It would be good to show genes/gene families/SNP differences between environmental isolates and clinical isolates in a figure (it should be a main figure).

b. Given that authors have identify clades of environmental, animal and clinical isolates I think authors should conduct positive selection analyses to investigate if there are certain genes under positive selection when *A. baumannii* become a hospital strain. Moreover, this will allow you to identify if there are particular genes that show converged positive selection when strains become clinical isolates across the clades.

c. Molecular clock analysis: What *Acinetobacter* species did the authors used as the outgroup when doing this analysis? Also given the diversity and the geographic distribution of the genomes you have, you should conduct ancestral state reconstruction to see if you can point our a potential origin of the radiation of this species clade.

We thank the reviewer for these important comments regarding the data we analyzed. We agree that throughout this study we only scratched the surface of the rich data sets but feel like the presented results will have a major contribution on future research within this field and that it is time to involve the scientific community to make full benefit out of these unique data. To comply with your suggestion (a), we performed gene family enrichment analyses (see novel Fig. 6 and novel Suppl. Figure 21). We are especially thankful for this suggestion given that it revealed a number of significant differences between environmental and clinical isolates' genomes which we are now presenting in Fig. 6 and commenting in lines 392-398.

(b) Positive selection analyses are beyond the scope of this study but will be surely performed by members of the scientific community soon and we would happily support such studies.

(c) See comments on the molecular clock analysis above in response to reviewer 1. We need to expand our dataset until we reach a better temporal signal. We followed the work by Holt et al. (2016).

4. Tracing the food sources: I think the proper way to do this is first to conduct metabarcoding of the pallet material. Then identify food items common in pallets with *A. baumannii* isolates and then collect the wild individuals (like what you did with earthworms) and then see if those wild diet items carry *A. baumannii*.

Definitely, this would have been an elegant way that will however be applied in the future to complete the tracing studies.

5. I also wonder why authors did not conduct whole microbiome sequencing of their samples to confirm that isolation actually represent the presence of *A. baumannii*. This is because the isolation alone might not give the full picture of the presence of *A. baumannii* in a given sample. Have you done any preliminary work to make sure you always identify *A. baumannii* from samples when the bacteria are present?

We agree, metagenomics studies are an important next step, however beyond the scope of this work. As already mentioned in the “Limitations” section of the Discussion, we have published evidence that we are not isolating *A. baumannii* from every sample that contains it. This has to be addressed in the future to complete our picture.

Minor comments

L127: Citations are weird. Please double check.

References were updated and included in the reference list.

L154: Is “average rate of 25%” mean they were isolated from 25% of samples. In that case isn't it the prevalence?

Correct! We rephrased accordingly.

L164: Terminology for birds is “Cloacal” not “Rectal”.

Corrected throughout!

L233: Can you split the isolate numbers of the bacteria according to different earthworm genera?

We have now provided this information in Supplementary Table S1.

L254-256: I think this is an issue of sampling depths and sampling types.

We have acknowledged this, now stating “...apparently differ from white stork regarding carriage of *A. baumannii* although this might be influenced by differences in sampling depth and sampling types.” (see lines 250-252).

Oxa-51 variants: What is your cutoff value to call a strain as a different variant? Is it 1bp difference? Also, what is the length of this segment?

Variants are only indexed if differing on the protein level, synonymous nucleotide substitutions justify no variant designation (see lines 191-193). The protein has a length of 274 amino acids.

L330-331: Can you summaries the geographic distribution of your genomes? This will help readers to easily see if there are geographic biases in the isolates you used in this study.

The geographical distribution of all genomes included (those selected from NCBI and ours) is available in Suppl. Table S4. The geographical distribution of the genomes generated within this study is highly biased (mainly from Poland and Germany) and accessible via the Microreact tool and illustrated in Fig. 4. However, the NCBI dataset is also highly biased: most genomes deposited have been contributed by a few countries worldwide.

L413: Is the incubation temperature you used natural (e.g., environmental temperature)? Also, if the adherence of *A. baumannii* inhibit the germination of fungal spores, can that also impact other properties of the spores (like the dispersal ability)?

Standard incubation condition was at 37°C, both for adherence and germination assays (we have now included this information in the Materials and Methods section (see also below), relevant at least for the compost setting. However, adherence and germination assays also work at 27°C. We don't know how adherence and dispersal interfere. We have refrained from such experiments because of biosafety concerns.

441-461: Very weird section to start your discussion with, given that the study was not a comparative study between *P. aeruginosa* and *A. baumannii*.

We have omitted this section on the comparison between *P. aeruginosa* and *A. baumannii*, also in agreement with other reviewers' recommendation.

462-501: This part is also very speculative, and I struggle to find how this relate to authors findings, except for that authors show the ability of the bacteria to adhere to fungi. I think if authors want to discuss more "Evidence for interrelatedness of *A. baumannii* life cycle with the fungal world" they need to conduct more experiment to properly show the claims they make here.

To extend the work on the interrelatedness of *A. baumannii* with fungi is beyond the scope of this study. Therefore, we have toned down on the evidence that we provide to comply with the reviewers' doubts. Moreover, we rewrote parts of the discussion and we updated literature in Suppl. Table S8. More and more evidence is accumulating in the literature but the topic is still clearly understudied. Given the high impact this has on the clinical outcome, we felt unable to completely omit this part. In particular, we believe awareness and research need to be stimulated.

L509: Basidiomycota is a very broad and diverse division of fungi and don't think using this is a good argument here.

We have specified "such as *Fistulina hepatica*" (see line 511).

L541-585: I am not sure how your findings support *A. baumannii* uses air currents to travel 1000s of kms between continents. Maybe I missed something here?

Thank you! This question helped us to optimize the presentation of our line of argument: We identified several isolates from comparably pristine habitats in Poland that are closely related to clinical isolates described in countries all over the world (e.g. Canada, Thailand, Japan). These pairs of environmental & clinical isolates each share most recent common ancestors that existed only a few decades ago. In comparison to bacteria that are known to spread in association with humans and which exhibit the highest dispersion rates (Louca, 2022), our findings suggest significantly higher dispersion rates for *A. baumannii*. How can we explain multiple independent long-distance spreading events within decades? Humans are not regularly colonized with *A. baumannii*. The genomes of our isolates show no signs of escape from hospitals or livestock production (e.g. in terms of acquisition of resistance determinants and mobile genetic elements). Accordingly, there is no evidence of spread due to human activity. *A. baumannii* is known to have the capability of airborne spread in hospitals and we have demonstrated repeatedly airborne colonization of sterile plant material in the garden. Aerosolization of *Acinetobacter* is in full agreement with the literature both in terms of metagenomics studies and in terms of resistance to desiccation and radiation. Especially striking, *A. baumannii* has been detected in glacier microbiomes all over the world including Antarctica. Moreover, a very recent paper by Rodó et al. (PMID: 39250672) explicitly reports on the possibility of long-distance transport of pathogens including *A. baumannii* based on tropospheric sampling. Arguably, we have no direct evidence that *A. baumannii* reaches jet streams, although not unlikely, and therefore removed allusions to this picture (see modification of title).

L559-560: This I don't understand. There seem to be an association with your isolates and presence of humans and then here you say "All in all, there is no evidence that the observed global pattern of distribution of environmental *A. baumannii* is caused by human activity". You have data to test this!

Thank you again! To clarify, human activity since the Neolithic generated numerous novel habitats including pastures and nutrient-rich waters while reducing non-conducive forest areas. This fueled early-on radiation within the species leading to a high diversity of ancient lineages. Environmental *A. baumannii* isolated in our study exhibit little to no signs of selection pressure as exerted by nowadays human activity, use of antibiotics in human health care and livestock production in particular. This has already been tested and presented in Suppl. Table S4 and Suppl. Figures S18 & S19 (see also modifications in the text in lines 380-384). Thus, there is no contradiction, ancient and modern human activity have different impact on the evolution of *A. baumannii*.

L660-668: This section seems like it is repeating things. Please add this section to other corresponding sections.

Done!

L809: Did the incubation happened at room temperature?

Incubation was at 37°C, which has now been stated in the manuscript (see lines 753-760).

L822-828: Results for this methods section is not in the "Results" section. I think it is very interesting and should be in the main paper.

Thank you! We agree that these results are very interesting, however due to the request to streamline and reduce the manuscript, we had to decide to keep it as it is; at least several aspects are provided in the Supplements (see Suppl. Fig. 20 and Suppl. Material S1).

Figures: Please provide better legends for the figures. The interactive link is cool but you need to provide enough information these figure legends to guide reads (figures and legends should be able to stand alone). Also please provide information on the color codes associated with phylogenies.

We provide developed figure legends of the figures to improve accessibility.

Point-by-point response:

Reviewer #4 (Remarks to the Author):

This study by Wilharm et al., provides an interesting dataset on *A. baumannii*, but I do not think authors are doing enough justification for this amazing dataset they have (i.e, genomes) produced here. This version of the manuscript still provides a simple story on where and how to isolate these bacteria (which is very interesting) and present some hypothesis on how this bacteria species might spread around the world. However, I still think the major value of this study are the new genomes, thus, I still think more comparative analyses of these genomes are necessary to properly understand the underlying genomic features that might drive *A. baumannii* to become clinically relevant strains (which should also play the center role).

Thank you for acknowledging our data collection. We agree in the need for more detailed comparative genomic analyses to better understand the particular success of a few lineages in hospitals as compared to the huge diversity found in the environment. However, such analyses require the isolation of IC 1-, IC 2- and IC 3-related strains from pristine environments which we could not achieve here (we already pronounced this in the manuscript in lines 415-417). This is especially important regarding IC 2, which is by far the most successful lineage in hospitals worldwide. Among the reasons, why we failed to isolate IC 2, the most plausible explanations are that IC 2 isolates (i) do not propagate in soil and white storks where most of our collection comes from, and/or (ii) their natural habitats are geographically restricted (see below), and/or (iii) they don't spread efficiently outside of hospitals (especially no efficient airborne spread). Therefore we believe this manuscript needs to be published: to stimulate further environmental studies all over the world to complete our picture!

Given that sampling is bias toward Germany and Poland, I don't think (even with the great number of isolates produce by this study), authors have enough evidence/data to give a global overview of this species complex. What you have are amazing set of genomes that you can answer many more interesting questions related to how *A. baumannii* become virulent in hospital settings. Such a story will fit well with the aims and interest the broad readership of this journal.

We agree that our collection does not give a global overview of the species complex. However, we would like to emphasize that we did not claim to have provided a global overview. On the contrary, we stated in the "Limitations" section (lines 588 ff) that there is evidence of understudying because of the use of a specific enrichment method and that the ecology of *A. baumannii* may differ in other climates and geographic regions. What is more, as outlined above, we missed to isolate some very important lineages (IC 1 -3) and have also clearly stated this. Nevertheless, the fact that our local sample set was able to represent more than half of the hitherto known diversity in terms of OXA-51 variants and to more than double the pangenome of the species is clear evidence of the global relevance of our study, let alone the close relationship of globally distributed isolates that we could demonstrate.

Secondly, I still don't agree with authors proposed mechanism of the spread of this species complex, mainly because the proposed mechanism is based on many non-tested assumptions or over extrapolating some findings. Also, authors tend to ignore potential steppingstone colonization methods, spread by other migrant species (not just storks), and potential combination of many methods including the airborne spread as authors suggested. I think, to receive a better outcome

from this amazing work, the scope of the study should change and generate more specific research questions (related to the evolution, and genetic underpinnings of virulence) instead of just saying you are trying to “improve the understanding of the ecology of *A. baumannii* in natural habitats...”.

We do not exclude that a combination of different mechanisms contributes to spreading of the species. Importantly, we do not even claim that storks contribute significantly to global spread of *A. baumannii*. However, our claim of long-distance airborne spread of *A. baumannii* is in full agreement with recent work by others as discussed in our manuscript, describing the presence of *A. baumannii* in glaciers worldwide and in tropospheric samples. Our trials to “improve the understanding of the ecology of *A. baumannii* in natural habitats...” turned *A. baumannii* from a nosocomial pathogen with unclear origin into one of the few bacterial pathogens really accessible for studies in its natural environments.